# Synthesis and Electrochemical Study of Three-Dimensional Graphene-Based Nanomaterials for Energy Applications

**DOI:** 10.3390/nano10071295

**Published:** 2020-07-01

**Authors:** Antony R. Thiruppathi, Boopathi Sidhureddy, Emmanuel Boateng, Dmitriy V. Soldatov, Aicheng Chen

**Affiliations:** Department of Chemistry, University of Guelph, Guelph, ON N1G 2W1, Canada; athirupp@uoguelph.ca (A.R.T.); bsidhure@uoguelph.ca (B.S.); eboate01@uoguelph.ca (E.B.); soldatov@uoguelph.ca (D.V.S.)

**Keywords:** soft nanomaterials, 3D nanostructure, graphene-based nanomaterials, supercapacitor, oxygen reduction

## Abstract

Graphene is an attractive soft material for various applications due to its unique and exclusive properties. The processing and preservation of 2D graphene at large scales is challenging due to its inherent propensity for layer restacking. Three-dimensional graphene-based nanomaterials (3D-GNMs) preserve their structures while improving processability along with providing enhanced characteristics, which exhibit some notable advantages over 2D graphene. This feature article presents recent trends in the fabrication and characterization of 3D-GNMs toward the study of their morphologies, structures, functional groups, and chemical compositions using scanning electron microscopy, X-ray diffraction, Raman spectroscopy, Fourier transform infrared spectroscopy, and X-ray photoelectron spectroscopy. Owing to the attractive properties of 3D-GNMs, which include high surface areas, porous structures, improved electrical conductivity, high mechanical strength, and robust structures, they have generated tremendous interest for various applications such as energy storage, sensors, and energy conversion. This article summarizes the most recent advances in electrochemical applications of 3D-GNMs, pertaining to energy storage, where they can serve as supercapacitor electrode materials and energy conversion as oxygen reduction reaction catalysts, along with an outlook.

## 1. Introduction

Graphene is a 2D material that possesses sp^2^ hybridized carbon atoms in a hexagonal lattice, which may be considered as single-layer graphite. It has been demonstrated that graphene is one of the thinnest and strongest materials known, having excellent flexibility and transparency, which also exhibits exceptional electrical and thermal conductivity, charge carrier mobility, as well as an extensive surface area [1,2]. Its properties may vary with respect to its structure, defect density, number of layers, etc. [3]. Due to these attributes, a number of graphene-based nanomaterials have been developed for utilization in energy storage, catalysis, electronic devices, and sensor applications [4,5,6,7,8,9]. Recently, graphene-based nanomaterials have also been utilized for aluminum ion batteries as a promising electrode material along with Al-foil in an ionic liquid electrolyte [10,11,12,13]. Intense efforts have been invested toward utilizing graphene to the fullest extent for a variety of practical applications. In general, 2D graphene-based nanomaterials are derived through chemical, electrochemical, mechanical, and solution-based approaches [14,15,16,17,18].

Various nanostructures can also be derived from 2D graphene. In its 2D form, graphene has a tendency to reaggregate due to weak van der Waals forces that can link its layers, which can alter its performance [19,20]. To overcome the aggregation issues, various strategies have been proposed toward the development of macroscopic 3D structures—for instance, the introduction of functional groups or spacers [21], use of interconnecting graphene sheets [15], and the deposition of 3D architectures [22]. Recently, 3D graphene-based architectures such as hydrogels [23,24], aerogels [25,26], foams [27,28], and sponges [29] have been reported in the literature. Apart from structural stability, these 3D architectures exhibited high electrical conductivity and improved heterogeneous electron transfer properties due to the interconnectivity of graphene sheets [30]. These 3D structures minimize the re-stacking effects that are commonly observed in 2D graphene-based materials, thus increasing the surface area, porosity, and pore volumes. Three-dimensional graphene-based nanomaterials (3D-GNMs) have been explored for their use as supercapacitors, in Zn-air batteries as electrode materials, in fuel cells as electrocatalysts or supports, as sensors, and in environmental remediation applications [31,32,33,34].

Electrochemical capacitors are electrochemical energy storage devices that bridge the gap between conventional electrolytic capacitors and batteries. Supercapacitors could store more energy than conventional capacitors. Unlike batteries, supercapacitors have their charge storage not limited by the diffusion of ions within bulk electrodes; thus, they can deliver high power and rapid charge/discharge cycles through ion adsorption and desorption [35]. Typically, electrochemical capacitors store energy via either the electrical double-layer or the pseudo-capacitance charge mechanism. For electrical double-layer capacitance, controlling the pore architecture of the materials is crucial for improved performance, where the morphology of materials plays a vital role. Whereas, for pseudo-capacitance, the functionalization of carbon-based nanomaterials with electroactive species is essential for the fast electron transfer reactions accompanied by high energy storage. Thus, the morphology and functionalization of a material play some important roles in advancing its supercapacitor application [36,37,38,39]. Along with desired morphologies, 3D materials possess a high ion accessible surface area, minimum inter-sheet resistance, and appropriate pore dimensions to deliver excellent performance for energy storage applications with high specific capacitance, rate capability, and cyclic stability [40,41].

The oxygen reduction reaction (ORR) plays a vital role in many electrochemical energy conversion and storage devices, such as fuel cells and metal–air batteries. The sluggish kinetics of ORR is a limiting factor of device performance; hence, it is essential to develop a stable, high-performance electrocatalyst for improving the efficiency and reducing the cost of devices [42,43]. Although advanced Pt/C electrocatalysts are available for the ORR, the commercialization of the technology is hindered due to the limitations of commercial catalysts, which include high cost, scarcity, and electrode poisoning in the presence of carbon monoxide and methanol. Graphene-based materials were shown to possess the catalytic ability for the ORR, which is attributed to the edges, defect sites, oxygen functional groups, and heteroatoms present in the structure [44,45]. Further, graphene-based nanomaterials have been explored as a catalyst support due to synergistic interactions between catalytic metal nanoparticles and metal oxides with graphene supports to improve catalytic abilities. The 3D morphology of graphene revealed effective mass transfer due to its high surface area as well as accessibility to a greater number of exposed planes. Various metal nanoparticles and complexes that decorated 3D graphene-based nanomaterials have been reported as high-performance catalysts [46]. The functionalization of graphene has been explored toward influencing its electronic and chemical properties, which can alter the dispersion, orientation, interaction, and electronic properties of 3D-GNMs. Various types of nanomaterials such as metal nanoparticles, metal oxides, heteroatoms, and conducting polymers have been explored for their potential synergies with 3D graphene supports to improve electrochemical energy storage and conversion performance [47,48,49,50,51,52,53,54]. Nitrogen is a well explored heteroatom that can enhance electroactive surface areas, improve conductivity, and improve wettability [55]. In addition, N doping induces ORR catalytic ability to graphene supports [56]. N-doping on 3D graphene has been explored through various routes, namely hydrothermal, solvothermal, thermal treatment, etc. [31,57,58]. The deposition of small quantities of metal nanoparticles such as Pd on graphene can enhance its ORR activities [59,60]. Metal nanoparticles or metal oxide-anchored 3D graphene hybrid nanocomposites can be synthesized through various strategies such as chemical reduction, hydrothermal, microwave irradiation, etc. [61,62,63].

In this feature article, the main strategies for the fabrication of 3D graphene-based nanomaterials are presented. The characterization of the morphologies, structures, and compositions of the fabricated nanomaterials using scanning electron microscopy (SEM) images, X-ray diffraction (XRD), Raman spectroscopy, Fourier transform infrared (FTIR) spectroscopy, and X-ray photoelectron spectroscopy (XPS) is addressed. The synthesis and characterization of the novel 3D graphene-based nanomaterials including interconnected reduced graphene oxide (IC-rGO), N-doped IC-rGO, and Pd-decorated N-IC-rGO as well as their promising applications in energy conversion and storage are highlighted. An outlook of the development of advanced 3D graphene-based soft nanomaterials for energy and environmental applications is discussed.

## 2. Fabrication of 3D Graphene-Based Nanomaterials

The synthesis and assembly of 3D graphene-based nanomaterials provide compelling research, particularly related to the control of their morphology and properties for practical applications. In this section, we will focus on the fabrication of 3D graphene architectures. As illustrated in Scheme 1, the fabrication process may be classified into three main types: (1) Template-Assisted Chemical Vapor Deposition Method; (2) Template-Assisted Chemical Method; and (3) Template-Free Chemical Method.

### 2.1. Template-Assisted Chemical Vapor Deposition Method

Chemical vapor deposition (CVD) is a well-established process for the synthesis of various carbon-based nanostructures, including carbon nanotubes, graphene, and other fullerenes, etc. [64]. Cheng et al. first introduced CVD for the synthesis of flexible three-dimensional interconnected graphene [65]. The basic strategy involved with this methodology was to initially produce a layer of graphene on a 3D foam-like structure followed by polymerization and etching, which led to 3D foam-like interconnected graphene networks. In general, Ni foam, Cu, ZnO, aluminium oxide, and pyrolyzed photoresist films have been incorporated as scaffolds and catalysts in the development of 3D nanostructures, while flat metal substrates were employed in the conventional methods for graphene growth [66,67,68]. In general, methane or acetylene gas is employed as a carbon source to grow the graphene on the substrate at high temperatures in the presence of hydrogen and argon. In the second step, the formed 3D scaffolds are coated with polymethyl methacrylate (PMMA). Finally, the Ni foam and polymer coating is dissolved, and the monolithic graphene 3D network is obtained [65]. Through the selection of appropriate scaffolds (Ni or Cu foam), the pore size and structures can be controlled in the resulting 3D interconnected graphene films. This template-assisted CVD technique has been demonstrated as a promising method for the fabrication of 3D graphene with excellent properties. However, the structural stability of these 3D monoliths still needs to be improved, as the interconnections between the graphene sheets are purely van der Walls forces.

### 2.2. Template-Assisted Chemical Method

Freestanding 3D graphene superstructures can also be prepared using a template-assisted chemical method. This process is similar to the above-discussed methodology; however, the starting materials are different. In general, the functionalized graphene is used as a precursor for this process, particularly graphene oxides and reduced graphene oxides [69]. The functionalized graphene is deposited either on polymer beads or metal foams after which a chemical or electrochemical process is performed for the reduction of functional groups [70]. Finally, the polymers or metal foams are dissolved in appropriate solutions to achieve 3D graphene monoliths [69,70]. Typically, polystyrene beads, Ni foam, Cu foam, CaCO_3_, and MnO_2_ have been employed as scaffolds [71]. In some cases, the potential application of 3D graphene with the scaffolds can also be established for supercapacitors and batteries [21,72,73]. The template-assisted chemical method has been widely used to develop 3D graphene architectures, where multiple steps and chemicals are involved for the fabrication of 3D graphene; however, this method needs to be simplified.

### 2.3. Template-Free Chemical Method

The Template-Free Chemical Method is also referred to as the chemical self-assembly method [70]. Three-dimensional scaffolds comprised of GO have been reported, with some of them being hydrogels and self-assembled films/aerogels [74,75,76]. This process can be subclassified into two types based on the interactions established between graphene (whether physical interactions and/or cross-linking agents enable contact between the graphene sheets). Most of these GO-based solids are obtained through the physical interactions of GO building blocks, and they depend on van der Waals interactions and the balanced electrostatic repulsive forces of GO functional groups such as hydroxyl, epoxy, and carboxyl groups [71]. Following the gelation or chemical reduction process, a special drying process (lyophilization) is required to obtain macroscopic 3D structures [69,70]. However, the mechanical stability of the resulting structure is poor without the inclusion of polymers or other organics. To overcome these issues, the interconnection of graphene sheets with covalent linkages was proposed and attempted with various network linkers such as glutaraldehyde, resorcinol, polyallylamine, DNA, etc. [23,77]. In some cases, these network linkers are electrochemically active; thus, their impacts on electrochemical applications are inevitable.

Our research group developed an innovative one-pot approach for the synthesis of interconnected reduced graphene oxide (IC-rGO) without the addition of cross-linked molecules, which we referred as a Streamlined Hummers Method (SHM) [78]. As shown in Scheme 2, IC-rGO was synthesized using the following two steps. In the initial step, 2.0 g of graphite and 200 mL of a 9:1 v/v proportion of sulfuric acid and phosphoric acid were mixed and stirred for 2 h at 50 °C. Subsequently, 9.0 g of potassium permanganate was gradually added, and the reaction was proceeded with stirring for an additional 15 h. Following that, the reaction mixture was cooled to 10 °C in an ice bath, after which 2.5 mL of 30% hydrogen peroxide was poured into the cold reaction mixture. Then, a fraction of the reaction mixture was separated from the first stage of the process to obtain GO. In the next step, the remaining reaction mixture was heated to 120 °C for 15 min to form the IC-rGO. The as-prepared GO and IC-rGO were washed with pure water, HCl, and ethanol, coagulated in diethyl ether, and then dried in a hot air oven at 50 °C overnight. Despite the long-existing method being available in the literature for the oxidation of graphite [79,80], most of the experiments were conducted under 90 °C. In our case, the reaction was carried at two different temperatures, where 50 °C was maintained for the first step, which was followed by the reaction temperature being increased to 120 °C in the second step. As GO possesses –OH and –COOH functional groups, the acid-catalyzed condensation process can spontaneously occur at 120 °C during the second step, which enables covalent linkages between graphene sheets via the formation of ether and ester groups. Our results revealed that a significant quantity of oxygen functional groups was removed from the GO that was formed during the first step, and the interconnection of the GO sheets was simultaneously enabled through an acid-catalyzed condensation process without the addition of external linkers by simply increasing the temperature from 50 to 120 °C.

The structural stability of the formed IC-rGO was impressive due to existing covalent bonds between the graphene sheets. We also attempted to modify the IC-rGO with various dopants such as nitrogen and palladium [81]. The N-doping of IC-rGO was accomplished using a hydrothermal method, where ammonium fluoride was used as the N precursor. In short, IC-rGO (2.0 mg mL^−1^) is dispersed in water and sonicated for 30 min. Then, ammonium fluoride (2 M) was poured into the dispersion while stirring; after that, the mixture was transferred to a 40 mL Teflon vessel and kept in an autoclave to be treated at 180 °C for 6 h. After cooling to room temperature, the obtained N- IC-rGO was carefully washed with pure water and ethanol, and then, it was dried in a hot air oven overnight. To further modify the N-IC-rGO, Pd nanoparticles were deposited using the soft alcohol reduction (SAR) method [82], where the estimated amount (for 10 wt % Pd loading) of palladium nitrate salt precursor was reduced and decorated onto the N-IC-rGO using an ethanol/water (1:1) mixture. After being washed with water and ethanol, the obtained solid product was dried in a hot air oven overnight. In all cases, namely IC-rGO, N-IC-rGO, and Pd-N-IC-rGO, the 3D structure was retained, and our studies confirmed the role of interconnectivity in the formed 3D nanomaterials.

## 3. Surface Characterization of 3D Graphene-Based Nanomaterials

### 3.1. Morphology

As 3D-GNMs were synthesized from graphite, the morphological transformation was examined using Scanning Electron Microscopy (SEM) (FEI Quanta FEG 250). Figure 11A–D depict SEM images of graphite, GO, IC-rGO, and N-IC-rGO. In Figure 1A, graphite shows a flake-like morphology, which is an indication of its crystallinity [83]. The morphology of GO shown in Figure 1B is formed following the complete oxidation of graphite at the end of the first step, which reveals a crumpled sheet-like structure [81]. These creases in the structure are formed as a result of the functionalization and exfoliation of graphite. Figure 1C displays the interconnected 3D morphology of IC-rGO [78]. The functionalized 2D sheets served as building blocks, which were interconnected through functional groups to form a 3D structure. Figure 1D reveals the morphology of N-IC-rGO [81], confirming that the interconnected 3D morphology was preserved following the N-doping process.

### 3.2. Structural Characterization

X-ray diffraction (XRD) is one of the most valuable techniques that is extensively used for the characterization of carbon-based materials including carbon nanotubes, diamond, activated carbon, as well as 2D and 3D graphene [84,85]. As a non-destructive technique, XRD is used to determine interlayer spacings, elucidate structural strains, and detect impurities. Particularly with graphene-based materials, XRD analyses are employed to assess the effectiveness of oxidation, expansion of graphite, and reduction of graphene oxide. All XRD measurements were carried out using a powder X-ray diffractometer (PANalytical Empyrean) with Cu Kα radiation (λ = 1.54184 Å) and PIXcel^1D^ linear detector. Typically, X-ray diffractograms of graphite exhibit two characteristic peaks at 2θ = 26° and 43°, corresponding to (002) and (100) plane reflections, respectively (JCPDS No. 01-0646) [86], which is shown as an example in Figure 2A. Upon the oxidation of graphite to form GO, the same reflection (002) is typically observed; however, it is shifted toward a lower angle at 2θ = 11°. The interlayer distance corresponding to the (002) peak increases to 0.794 nm from 0.336 nm following the insertion of oxygenated functional groups, which confirms the efficacy of the oxidation reaction. The reduction of GO to rGO may be achieved through the chemical or thermal reduction method. The XRD pattern for IC-rGO (3D graphene) possesses a similar pattern associated with rGO (2D graphene) [78]. Upon the reduction of GO to IC-rGO followed by N-doping, it was noted that the (002) plane reflection of the GO moved toward a higher angle for both IC-rGO and N-IC-rGO. Therefore, smaller d-spacing was observed for IC-rGO (0.355 nm) and N-IC-rGO (0.351 nm). The decreased d-spacing upon reduction indicated an efficient reduction of the oxygenated functional groups in the GO. Furthermore, a peak broadening observed at 2θ = approximately 24° indicated that the mean crystallite lengths along the c-axes (Lc) of both IC-rGO (1.8 nm) and N-IC-rGO (1.4 nm) were shorter than that of GO (6.4 nm). Using the calculated Lc and interlayer distance, the number of graphene layers in a stacked nanostructure can be estimated to be approximately 8, 5, and 4 for the formed GO, IC-rGO, and N-IC-rGO, respectively.

Raman spectroscopy is employed extensively for the structural and electronic characterization of graphitic materials. Raman spectra provide some useful information on defects, carbon sp^2^ vibrations, and stacking orders [87]. All Raman spectra were recorded on a Renishaw micro-Raman spectroscope with an excitation laser beam wavelength of 532 nm. Generally, it is found that most carbon-based nanomaterials, including graphene, present almost identical Raman peaks comprised of characteristic D, G, and 2D modes at approximately 1352, 1580, and 2721 cm ^−1^, respectively. Graphite without disorder displays only the G band at approximately 1580 cm^−1^. When disorder is introduced within graphite, a defect-induced (D) band appears at approximately 1350 cm^−1^ [87]. For example, the Raman spectra of graphene-based materials, including graphite, GO, 3D IC-rGO, and 3D functionalized graphene nanocomposites such as N-IC-rGO are presented in Figure 2B, confirming almost identical peaks in the presence of D and G bands. A defect density in the graphene structure may be obtained by comparing the intensity ratio of the D and G bands. The Raman spectra obtained for graphite, GO, 3D IC-rGO, and N-IC-rGO displayed a defect density of 0.05, 0.89, 0.84, and 0.80, respectively. Graphite has the smallest defect density; the introduction of oxy-containing functional groups due to the chemical oxidation resulted in the increase of the disorder and defect density of GO. After reduction, a decrease of the defect density was observed due to the partial removal of the oxy-containing functional groups, which is in agreement with the XRD results. Raman spectroscopy may also be used to verify the in-plane crystallite sizes (La) of the graphene samples by employing the La = 4.4 (I_G_/I_D_) relation, as the in-plane crystallite size is inversely proportional to the defect density; thus, smaller defect density corresponds to a larger crystallite size and vice versa.

Fourier transform infrared (FTIR) spectroscopy is a useful technique for qualitatively identifying surface functional groups that are present in graphene-based nanomaterials. FTIR spectra were recorded using a Thermo Scientific spectrometer. An example is demonstrated in Figure 3, which displays the FTIR spectra of GO, 2D T-rGO, and 3D IC-rGO. In the obtained spectra, the existence of characteristic peaks at 3000–3500 cm^−1^ (hydroxyl, OH group), 1730 cm^−1^ (carbonyl, C=O group), 1610 cm^−1^ (C=C), 1226 cm^−1^ (epoxy), and 1049 cm^−1^ (alkoxy) were evidenced for GO. However, upon reduction by a thermal method, the formation of T-rGO and IC-rGO resulted in a weakened OH peak, which confirmed the efficient removal of oxygen functional groups. The extensive removal of oxygen functional groups during heat treatment under an inert environment results in 2D crumpled sheet-like morphologies. Compared with 2D T-rGO, the FTIR spectrum for 3D IC-rGO displayed a strong absorption peak at 1209 cm^−1^, which could be assigned to the formation of ethers and esters (C–O–C). The formation of ether and ester groups may play an essential role toward interconnecting functionalized GO sheets to form 3D IC-rGO. In N-IC-rGO, the C–N stretching bonds appeared at approximately 1100 and 1570 cm^−1^, which confirmed successful N-doping [88,89]. In addition, decreases in the intensity of C=O stretching bands indicated a further reduction of functional groups during the N-doping process. Furthermore, sustained C–O–C bands confirmed the preservation of interconnected 3D structures following N-doping.

The accurate chemical composition of 3D-GNMs was determined using XPS measurements, which were performed using a Thermo Scientific K-α XPS spectrometer. All prepared samples were run at a takeoff angle of 90°. A monochromatic Al Kα X-ray source was employed with a spot size of 400 μm. The charge compensation was measured, and the location of the energy scale was adjusted to place the main C1s feature (C–C) at 284.6 eV. All data analyses were carried out using XPS peakfit software. Figure 4A,C displays the XPS survey spectra, high-resolution C1s, and high-resolution N1s spectra of the 3D graphene-based nanomaterials. Figure 4A shows the XPS survey spectra and peaks labeled as C1s (approximately 285.0 eV), O1s (approximately 533.0 eV), N1s (approximately 399.5 eV), and S1s (approximately 171.0 eV). The C/O ratios for graphite, GO, IC-rGO, and N-IC-rGO were estimated to be 12.7, 1.9, 3.9, and 5.8, respectively. The C/O ratio decreased from graphite to GO, which indicated the introduction of oxygen functional groups. From GO to IC-rGO, a significant number of functional groups were reduced; thus, the C/O ratio decreased to 1.85. Furthermore, a small quantity of S (approximately 1.0 at %) observed in GO dissipated during the formation of IC-rGO. For N-IC-rGO, the C/O increased due to the further reduction of oxygen functional groups. In addition, from the N-IC-rGO’s survey spectra, the N-doping level was determined to be 4.74%.

A high-resolution C1s spectrum can be used to understand the type of functional groups of carbon-based nanomaterials. Figure 4B shows that the high-resolution C1s spectra of all four materials may be deconvoluted to five different C bonding configurations. These peaks might be labeled as C=C (284.6 eV), C–C (285.5 eV), C–O (286.6 eV), C=O (287.6 eV), and O=C–O (288.8 eV) [90]. The C1s spectrum of GO was significantly different from graphite, which indicated that the drastic change in the structure was due to the introduction of oxygen functional groups [4]. On comparing the C1s spectra of GO and IC-rGO, it was evident that the quantity of oxygen functional groups was significantly decreased [78]. The N-IC-rGO’s C1s spectrum shows the further reduction of C–O and C=O and the introduction of N containing functionalities, and C–N bonds were overlaid with C–O binding energies [81]. The proportion of functional groups is shown in Table 1.

Figure 4C displays the high-resolution N1s spectrum of N-IC-rGO, which can be deconvoluted to three N configurations, namely pyridinic N (398.5 eV), pyrrolic N (399.6 eV), and graphitic N (401.8 eV) [91], and their relative percentages were estimated to be 30%, 60%, and 10%, respectively. Pyrrolic N was predominantly formed in N-IC-rGO, which had increased activity at the edges and contributed to improving the electron donor behavior of the 3D graphene host and pseudo-capacitance [81].

## 4. Electrochemical Applications of 3D Graphene-Based Nanomaterials

The 3D graphene-based nanomaterials possess some unique properties and enable various environmental, energy, and sensing applications. For instance, supercapacitors are attractive devices for energy storage with a wide range of applications from portable electronics to electric vehicles. The fabricated 3D-GNMs were tested for supercapacitor applications, where a CHI 660E potentiostat was employed for all electrochemical experiments. A three-electrode electrochemical cell was utilized. The pre-determined active materials (IC-rGO, N-IC-rGO) were added to 950 μL of pure water and 50 μL of a 10% Nafion binder. After being ultrasonicated for 30 min, an aliquot (3.0 μL) of the homogenous dispersion was cast on a well-polished glassy carbon electrode (GCE) surface (area 0.07 cm^2^) and dried at room temperature overnight before the electrochemical measurements. A polycrystalline Pt wire and Ag/AgCl (saturated KCl) were used as the counter and reference electrodes, respectively.

### 4.1. Electrochemical Energy Storage Application

All the electrochemical experiments were performed in 0.5 M H_2_SO_4_. Cyclic voltammogram (CV) was used to understand the capacitance and the mechanism of energy storage. Specific capacitance could be estimated from the area of cyclic voltammogram. Figure 5A shows the CV of IC-rGO with GO and 2D thermally reduced graphene oxide (T-rGO). The rectangular shape of the CV combined with a hump-like feature indicated the contribution of non-Faradaic (arising from double layer), as well as the Faradaic capacitance (arising from residual oxygen functional groups). IC-rGO exhibited a capacitance of 158 F g^−1^, which was significantly higher compared to 2D T-rGO (54.2 F g^−1^) and GO (2.3 F g^−1^). As seen in the SEM images, the 3D interconnected structure of IC-rGO increased the active surface area and minimized restacking while processing as an electrode compared to the 2D T-rGO. Constant current charge/discharge studies are another useful technique to assess the specific capacitance. Here, the capacitance is proportional to the discharge duration. Figure 5B presents the constant current charge and discharge profile of IC-rGO and 2D T-rGO. The capacitance of IC-rGO was 212 F g^−1^ at 1 A g^−1^ and 176 F g^−1^ at 10 A g^−1^, which is double that of the capacitance calculated for 2D T-rGO, which is 93 F g^−1^ at 1 A g^−1^ and 70 F g^−1^ at 10 A g^−1^. Furthermore, the longevity of the energy storage material is assessed through cycle life testing. Cycle life testing is continuously performing charge/discharge at a specific current for a prolonged duration. Figure 5C shows the cycle life testing of IC-rGO at 10 A g^−1^ for 5000 cycles, and the insert shows the stable charge/discharge pattern at different stages of the testing procedure. It is evident that the capacity of IC-rGO was retained without any noticeable loss over 5000 cycles.

So far, the 3D IC-rGO demonstrated higher stability and higher capacitance compared to its 2D T-rGO counterpart. Next, we investigated the effect of N doping on the capacitance. Figure 6A–D present the results of N-IC-rGO as a supercapacitor electrode material. Figure 6A displays the CVs of the IC-rGO and N-IC-rGO at a 0.1 V s^−1^ scan rate. The calculated capacitance of N-IC-rGO was 327 F g^−1^, which was significantly higher compared to that of IC-rGO. Figure 6B shows the constant current charging/discharging performance of N-IC-rGO in contrast to IC-rGO. The capacitance determined from the charge/discharge studies was found to be 319 F g^−1^. The significant increase in capacitance was attributed to the introduction of N, which altered the electronic structure of the 3D IC-rGO.

The presence of pyrrolic N and pyridinic N is known to improve pseudo-capacitance, whereas graphitic N increases electronic conductivity. This excellent performance motivated us to investigate the performance in a symmetric two-electrode capacitor. Graphite rods were used as current collectors, polyvinyl alcohol–phosphoric acid gel was used as the electrolyte, and filter paper was used as a separator. Active material loading on the electrodes was approximately 1 mg cm ^2^. Figure 6C depicts the CV at different scan rates from 6 to 600 mV s^−1^. The shape of the CV pattern was consistent at different scan rates, which was indicative of rate capability—that is, retaining capacitance over fast charge/discharge conditions. N-doping not only increased the capacitance but also enhanced the rate capability. The improved rate capability was attributed to the smaller resistance, which was attributed to the introduction of graphitic N. Furthermore, the long-term performance stability of N-IC-rGO was examined through cycle life tests at 10 A g^−1^ charging and discharging. Figure 6D shows that the capacitance was stable over 2630 cycles, and the insert confirms that the charge/discharge pattern was consistent over the course of cycle life testing.

So far, the energy storage performance of the 3D-GNMs was demonstrated based on the IC-rGO and N-doped IC-rGO. Several other 3D graphene structures such as aerogels and hydrogels have also been explored for supercapacitor materials. Besides N-doping, the co-doping of heteroatoms (N and S) for energy storage has also been reported. Doping single heteroatom improves capacitance as a result of tuning the electronic structure of graphene, whereas dual heteroatoms provide unique features of both heteroatoms, which further enhances the energy storage performance. Furthermore, 3D-GNMs combined with metal oxides (MnO_2_), metal chalcogenides (MoS_2_), and conducting polymers (polyaniline) have also been reported as supercapacitor materials. 3D graphene facilitates the uniform distribution of the metal oxides/chalcogenides, minimizes the interfacial resistance, and improves the stability of the hybrid nanocomposites. The hierarchical structures of metal oxides/chalcogenides via their pseudo-capacitance sites increase the ion accessible area, which contributes to the energy density and stability of the energy storage device. Table 2 presents 3D-GNM materials with their preparation methods and doping levels as the corresponding energy storage performance with electrolyte, testing cell configuration, and specific capacitance.

### 4.2. Electrochemical Energy Conversion Applications

Energy conversion devices play a vital role in meeting the ever-growing global demands for inexhaustible and eco-friendly energy. In recent years, 3D-GNM have been widely explored as catalysts, which support economically important energy conversion reactions, such as the oxygen reduction reaction (ORR), water splitting, carbon dioxide reduction, etc. ORR is a cathodic reaction in fuel cells and air batteries. In this section, the catalytic activity of 3D-GNM is demonstrated using two examples—namely, N-IC-rGO and Pd-N-IC-rGO for oxygen reduction reaction.

#### 4.2.1. Oxygen Reduction Performance of N-IC-rGO

All measurements were performed using a rotating disc electrode (RDE) of 0.19 cm^2^ with active material loading 0.21 mg cm^−2^ in 0.1 M KOH electrolyte. The CV technique is useful to confirm the catalytic activity. Figure 7A,B depict the CVs of IC-rGO, N-IC-rGO, and N-rGO in argon-saturated and oxygen-saturated environments, respectively. As seen in Figure 7A, there was no observable peak, indicating that there is no activity in an argon-saturated electrolyte. In Figure 7B, there was an obvious reduction peak observed at the potential of approximately −0.25 V Vs Ag/AgCl for N-rGO and N-IC-rGO in oxygen saturated electrolyte, confirming the ORR catalytic activity. From visual comparison of the onset potential and peak currents corresponding to the reduction peak from the CV, N-IC-rGO exhibited significantly more positive onset potential and higher current compared to N-rGO and IC-rGO. Furthermore, the catalytic activity was probed using a linear sweep voltammogram (LSV) with rotation of the working electrode at specific revolutions per minute (RPM) to minimize the diffusion limitation. Figure 7C shows the LSVs of IC-rGO, N-IC-rGO, and N-rGO recorded at a scan rate of 0.002 V s^−1^ at 1600 RPM. Figure 7D illustrates a Tafel plot (η (overpotential) Vs log j (current density)), which was constructed using LSV data (Figure 7C). The estimated onset potentials for ORR from LSVs of N-IC-rGO, IC-rGO, and N-rGO were −0.05 V, −0.12 V, and 0.08 V, respectively. The limiting current and half-wave potential of N-IC-rGO were found to be −2.9 mA cm^−2^ and −0.18 V, respectively. Further, the Tafel slopes of N-IC-rGO, IC-rGO, and N-rGO were determined to be 80 mV, 128 mV, and 97 mV, respectively. The N-IC-rGO exhibited a significantly more positive onset potential and half-wave potential, increased the current density, and had the smallest Tafel slope in contrast to IC-rGO and N-rGO. This catalytic activity was entirely attributed to the N functionalities on the 3D structured IC-rGO. The presence of graphitic N improved the conductivity and electron density on the IC-rGO, which provided active sites for the ORR.

To understand catalyst selectivity, it is important to study the ORR reaction pathway. A set of LSVs recorded with different RPMs are shown in Figure 8A. As expected, increased RPM resulted in higher current densities. As shown in Figure 8B, a Koutecky–Levich (K-L) plot was constructed using the LSV data from Figure 8A. The number of electrons transferred during the ORR was estimated using the K-L equation [103]. According to the K-L equation, the measured current density is a combination of diffusion limited and kinetic limited current density (Equation (1)), where J_k_ = kinetic current density, J_d_ = diffusion current density, 1/B is the slope, and ω = angular rotation speed.
(1)$$\frac{1}{J}=\frac{1}{J_{k}}+\frac{1}{J_{d}}=\frac{1}{J_{k}}+\frac{1}{B\omega^{1/2}}$$

Figure 8B shows the K-L plots (J^−1^ Vs ω^−0.5^) constructed from the LSV data at different revolutions per minute (RPM). Using the slope (1/B), the number of transferred electrons was determined using Equation (2), where n = number of electrons, F = Faradaic constant, A = area of the electrode, D_(O2)_ = diffusion coefficient of oxygen gas, ϑ = kinetic viscosity of solution (0.01 cm^2^/s), and C_(O2)_ = concentration of dissolved oxygen (1.26 × 10^−3^ M).
(2)$$B=0.2nFA{\left(D_{O_{2}}\right)}^{2/3}\vartheta^{-1/6}C_{O_{2}}$$

From the above calculations the number of electrons transferred during the ORR on the N-IC-rGO was found to be 4.0, which is shown in the insert of Figure 8B. This confirmed that N-IC-rGO was highly selective for the reduction of oxygen to form water through four-electron transfer. Pyridinic N and graphitic N present in the 3D N-IC-rGO were the active sites responsible for the ORR activity, and this observation was consistent with the XPS results [104]. Therefore, this demonstrates that N-doping and 3D morphology synergy contributed to the higher ORR activity of N-IC-rGO compared to its 2D counterpart N-rGO.

#### 4.2.2. Oxygen Reduction Performance of Pd-N-IC-rGO

Furthermore, we investigated the effects of metal nanoparticle deposition on the 3D N-IC-rGO. Pt/C is a state-of-the-art catalyst for ORR; however, it is expensive and scarcely available. Therefore, we selected Pd, which is cheaper than Pt. We aimed to improve the mass activity of Pd by anchoring it on an active 3D N-IC-rGO catalyst. Figure 9A,C present ORR results for N-IC-rGO and Pd-N-IC-rGO.

Firstly, Pd decoration on N-IC-rGO was accomplished with an environment-friendly soft alcohol reduction method (SARM) [82]. The decoration of Pd on the 3D N-IC-rGO was confirmed by energy-dispersive spectroscopic (EDS) analysis showing that Pd content was 4.1 ± 0.1 wt %. Following the confirmation of Pd decoration, the ORR activity was probed using the RDE experiments. Figure 9A shows the LSV performed at 1600 RPM at a 0.005 V s^−1^ scan rate. The Pd-N-IC-rGO apparently showed improved ORR activity onset potential and limiting current density. The onset potential, half-wave potential, and limiting current density of Pd-N-IC-rGO from the LSV were 0.02 V, −0.16 V Vs Ag/AgCl, and 3.34 mA cm^−2^. Furthermore, the reaction pathway was studied using a K-L plot. Figure 9B shows a series of LSVs performed at different RPMs. Figure 9C shows the K-L plot constructed using the LSV data. The insert of Figure 9C shows that the number of electrons transferred was approximately 4.0, which was the same as for N-IC-rGO, further confirming the high catalytic activity and selectivity. The Pd decoration further improved the ORR activity of N-IC-rGO by positively shifting the onset potential and increasing the limiting current density. Besides the heteroatom doping and the modification of nanoparticles, various hybrid catalysts using 3D-GNM have been reported in the literature. Table 3 summarizes the ORR activity of nanocomposites with different 3D graphene supports, such as graphene foam, graphene aerogel, 3D graphene with dual heteroatoms (N&S, B&N), metal oxides (Mn_3_O_4_), mixed metal oxides (NiCo_2_O_4_, CoFe_2_O_4_), alloy nanoparticles (Ni-Fe), etc. N-doping on the 3D IC-rGO increased the ORR activity compared to 2D N-rGO. Pd nanoparticle decoration onto N-IC-rGO further improved the ORR activity.

## 5. Summary and Future Outlook

In this feature article, we initially discussed recent advances in the fabrication of 3D graphene-based nanomaterials. Subsequently, the characterization techniques used to study the 3D materials were discussed, using the IC-rGO and N-IC-rGO as models. Morphological transformations were characterized using SEM, whereas the structural changes and chemical compositions were elucidated using XRD, Raman, FTIR, and XPS techniques. The aggregation of 2D graphene sheets could potentially inhibit ion transport into its structure, which was overcome with 3D network macrostructures such as IC-rGO and N-IC-rGO. The advantages of the 3D morphology were elucidated through the electrochemical energy storage performance of IC-rGO, while the benefits of heteroatom doping onto 3D graphene were investigated through N-IC-rGO. N-doping on a 3D-GNM enhanced the specific capacitance and the rate capability. Moreover, N-doping on the 3D-GNM was shown to have catalytic activity for the ORR. In addition to the N-doping effect, we also demonstrated that the decoration onto the 3D N-IC-rGO with Pd nanoparticles further improved the ORR performance. Overall, this article summarized recent progress in the fabrication of 3D graphene hierarchical structures and their characterization, as well as their electrochemical energy storage and oxygen reduction catalysis applications.

The rising global demand for energy storage and production makes graphene-based nanomaterials important candidates for environmentally compatible energy storage and conversion devices. In its pure form, graphene has remarkable properties; however, it suffers limitations for large-scale production due to the re-stacking issues. This key challenge can be addressed through the development of 3D hierarchical structures. Various 3D-GNMs have been reported, including IC-rGO, hydrogel, aerogel, sponge, foam, etc. In addition, pore architectures (porous graphene, mesoporous/microporous materials) were evolved to improve the performance of graphene-based materials. Current trends involve the development of hybrid materials to exploit the synergies resulting from the combination of 3D-GNM with functional candidates, such as carbon materials (CNT), metal/alloy nanoparticles, metal oxides, metal hydroxides, metal sulfides, etc., for energy storage and catalytic applications. Based on this, possible future research directions include (1) exploration of novel materials considering the practical needs of device energy/power outputs such as supercapacitors, air batteries, and fuel cells; and (2) the development of promising materials for flexible and compact energy storage devices, which would advance a myriad of applications in wearable electronics and biomedicine. The outlook for 3D-GNMs is to identify beneficial synergies via a fundamental understanding of their formation mechanisms, which will lead to breakthroughs toward the development of custom materials for specific needs that can meet energy demand challenges. In view of the early improvements demonstrated here, we believe that the future potential of 3D graphene-based materials is immense.

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
