# Peer review of "Synthesis and Electrochemical Study of Three-Dimensional Graphene-Based Nanomaterials for Energy Applications"

_nanomaterials, 2020, doi:10.3390/nano10071295_

Round 1
Reviewer 1 Report
Authors introduce that recent synthesis and electrochemical Study of 3D Graphene-Based Nanomaterials for Energy Applications and provide s clear description on this topic. I suggest that it is suitable to be published on Nanomaterials with minor revision.
1. Graphene is very useful materials for energy storage application but they did not mentioned Al ion battery (AIB). Recently, AIB showed a good potential to be replace Lead acid battery because AIB was assembled by low cost materials (graphite or graphene and Al metal foil) with using ionic liquid electrolyte. Some references were worth to be mentioned as following: Nature Communicationsvolume 8, Article number: 14283 (2017), Carbon 146 (2019) 528e534, RSC Adv., 2019, 9, 11322, ACS Appl. Mater. Interfaces 2020, 12, 2, 2572-2580.
Author Response
Dear Reviewer:
Many thanks for reviewing our manuscript and for your very positive comments.
According to your suggestion, we have made the following changes:
- On Page 2, the following sentence "Recently, graphene-based nanomaterials has also been utilized for aluminum ion batteries as an promising electrode material along with Al-foil in an ionic liquid electrolyte [10–13]" has been added.
- Your suggested papers are interesting and have been cited as Ref. 10-13.
- All the changes have been tracked in the revised manuscript.
Best regards,
Aicheng Chen
Reviewer 2 Report
The review is well written, logically organized and satisfactorily covering the field. The review can be thus published in its present form.
Author Response
Dear Reviewer:
Many thanks for reviewing our manuscript and for your very positive comments.
Best regards,
Aicheng Chen
Reviewer 3 Report
The authors drafted a short review article that includes the most significant results on graphene-based three-dimensional nanomaterials for electrochemical applications. The first part of the document provides a description of the most effective pathways for obtaining complex graphene structures. Furthermore, the most used experimental techniques for obtaining structural and electronic information of the materials are described. The last part of the article illustrates some interesting results on electrochemical applications. I think the authors have done an excellent synthesis work on a very vast and constantly growing research field. The scientific data cited are correct and the bibliography allows to deepen the different topics. I think the article can be accepted in its current form.
Author Response
Dear Reviewer:
Many thanks for reviewing our manuscript and for your very positive comments.
Best regards,
Aicheng